# SYNTHETIC AND NATURAL NOISE BOTH BREAK NEURAL MACHINE TRANSLATION

**Yonatan Belinkov**[*]
Computer Science and
Artificial Intelligence Laboratory,
Massachusetts Institute of Technology
belinkov@mit.edu

**Yonatan Bisk**[*]
Paul G. Allen School
of Computer Science & Engineering,
University of Washington
ybisk@cs.washington.edu

## ABSTRACT

Character-based neural machine translation (NMT) models alleviate out-of-vocabulary issues, learn morphology, and move us closer to completely end-to-end translation systems. Unfortunately, they are also very brittle and easily falter when presented with noisy data. In this paper, we confront NMT models with synthetic and natural sources of noise. We find that state-of-the-art models fail to translate even moderately noisy texts that humans have no trouble comprehending. We explore two approaches to increase model robustness: structure-invariant word representations and robust training on noisy texts. We find that a model based on a character convolutional neural network is able to simultaneously learn representations robust to multiple kinds of noise.

## 1 INTRODUCTION

Humans have surprisingly robust language processing systems that can easily overcome typos, misspellings, and the complete omission of letters when reading (Rawlinson, 1976). A particularly extreme and comical exploitation of our robustness came years ago in the form of a popular meme:

> "Aoccdrnig to a rscheearch at Cmabrigde Uinervtisy, it deosn't mttaer in waht oredr the ltteers in a wrod are, the olny iprmoetnt tihng is taht the frist and lsat ltteer be at the rghit pclae."

A person's ability to read this text comes as no surprise to the psychology literature. Saberi & Perrott (1999) found that this robustness extends to audio as well. They experimented with playing parts of audio transcripts backwards and found that it did not affect comprehension. Rayner et al. (2006) found that in noisier settings reading comprehension only slowed by 11%. McCusker et al. (1981) found that the common case of swapping letters could often go unnoticed by the reader. The exact mechanisms and limitations of our understanding system are unknown. There is some evidence that we rely on word shape (Mayall et al., 1997), that we can switch between whole word recognition and piecing together words from letters (Reicher, 1969; Pelli et al., 2003), and there appears to be no evidence that the first and last letter positions are required to stay constant for comprehension.[1]

In stark contrast, neural machine translation (NMT) systems, despite their pervasive use, are immensely brittle. For instance, Google Translate produces the following unintelligible translation for a German version of the above meme:[2]

> "After being stubbornly defiant, it is clear to kenie Rlloe in which Reiehnfogle is advancing the boulders in a Wrot that is integral to Sahce, as the utterance and the lukewarm boorstbaen stmimt."

While typos and noise are not new to NLP, our systems are rarely trained to explicitly address them, as we instead hope that the relevant noise will occur in the training data.

Despite these weaknesses, the move to character-based NMT is important. It helps us tackle the long tailed distribution of out-of-vocabulary words in natural language, as well as reduce computation

---

[*]Equal contribution. Ordering determined by bartender's coin: https://youtu.be/BFSc2HnpYtA

[1]One caveat we feel is important to note is that most of the literature in psychology has focused on English.

[2]Retrieved on February 2, 2018.

load of dealing with large word embedding matrices. NMT models based on characters and other sub-word units are able to extract stem and morphological information to generalize to unseen words and conjugations. They perform very well in practice on a range of languages (Sennrich et al., 2016a; Wu et al., 2016). In many cases, these models actually discover an impressive amount of morphological information about a language (Belinkov et al., 2017a). Unfortunately, training (and testing) on clean data makes models brittle and, arguably, unfit for broad deployment.

Figure 1 shows how the performance of two state-of-the-art NMT systems degrades when translating German to English as a function of the percent of German words modified. Here we show three types of noise: 1) Random permutation of the word, 2) Swapping a pair of adjacent letters, and 3) Natural human errors. We discuss these types of noise and others in depth in section 4.2. The important thing to note is that even small amounts of noise lead to substantial drops in performance.

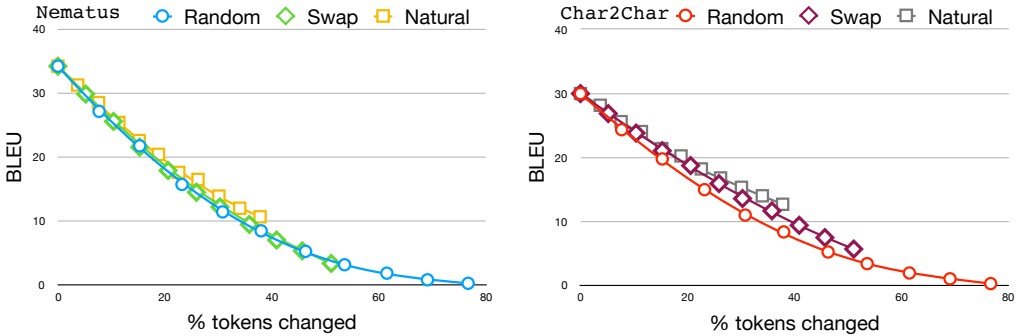

Figure 1: Degradation of `Nematus` (Sennrich et al., 2017) and `char2char` (Lee et al., 2017) performance as noise increases.

To address these trends and investigate the effects of noise on NMT, we explore two simple strategies for increasing model robustness: using structure-invariant representations and robust training on noisy data, a form of adversarial training (Szegedy et al., 2014; Goodfellow et al., 2015). We find that a character CNN representation trained on an ensemble of noise types is robust to all kinds of noise. We shed some light on the model ability to learn robust representations to multiple types of noise, and point to remaining difficulties in handling natural noise. Our goal is two fold: 1) initiate a conversation on robust training and modeling techniques in NMT, and 2) promote the creation of better and more linguistically accurate artificial noise to be applied to new languages and tasks.

## 2 ADVERSARIAL EXAMPLES

The growing literature on adversarial examples has demonstrated how dangerous it can be to use brittle machine learning systems so pervasively in the real world (Biggio et al., 2012; Szegedy et al., 2014; Goodfellow et al., 2015; Mei & Zhu, 2015). Small changes to the input can lead to dramatic failures of deep learning models (Szegedy et al., 2014; Goodfellow et al., 2015). In the machine vision field, changes to the input image that are indistinguishable by humans can lead to misclassification. This leads to potential for malicious attacks using adversarial examples. An important distinction is often drawn between white-box attacks, where adversarial examples are generated with access to the model parameters, and black-box attacks, where examples are generated without such access (Papernot et al., 2016a; 2017; Narodytska & Kasiviswanathan, 2017; Liu et al., 2017).

While more common in the vision domain, recent work has started exploring adversarial examples for NLP. A few white-box attacks have employed the fast gradient sign method (Goodfellow et al., 2015) or other techniques to find important text edit operations (Papernot et al., 2016b; Samanta & Mehta, 2017; Liang et al., 2017; Ebrahimi et al., 2017). Others have considered black-box adversarial examples for text classification (Gao et al., 2018) or NLP evaluation (Jia & Liang, 2017). Heigold et al. (2017) evaluated character-based models on several types of noise in morphological tagging and MT, and observed similar trends to our findings. Finally, Sakaguchi et al. (2017) designed a character-level recurrent neural network that can better handle the particular kind of noise present in the meme mentioned above by modeling spelling correction. Here we devise simple methods for generating adversarial examples for NMT. We do not assume any access to the NMT models' gradients, instead relying on synthetic and naturally occurring language errors to generate noise.

The other side of the coin is to improve models' robustness to adversarial examples (Globerson & Roweis, 2006; Cretu et al., 2008; Rubinstein et al., 2009; Chan et al., 2017). Adversarial training – including adversarial examples in the training data – can improve a model's ability to cope with such examples at test time (Szegedy et al., 2014; Goodfellow et al., 2015). This kind of defense is sensitive to the type of adversarial examples seen in training, but can be made more robust by *ensemble adversarial training* – training on examples transfered from multiple pre-trained models (Tramèr et al., 2017). We explore ensemble training by combining multiple types of noise at training time, and observe similar increased robustness in the machine translation scenario.

Training on and for adversarial noise is an important extension of earlier work on creating robustness in neural networks by incorporating noise to a network's representations, data, or gradients. Training with noise can provide a form of regularization (Bishop, 1995) and ensure the model is exposed to samples outside the training distribution (Matsuoka, 1992).

## 3  MT Systems

The rise of end-to-end models in neural machine translation has led to recent interest in understanding how these models operate. Several studies investigated the ability of such models to learn linguistic properties at morphological (Vylomova et al., 2016; Belinkov et al., 2017a; Dalvi et al., 2017), syntactic (Shi et al., 2016; Sennrich, 2017), and semantic levels (Belinkov et al., 2017b). The use of characters or other sub-word units emerges as an important component in these models. Our work complements previous studies by presenting such NMT systems with noisy examples and exploring methods for increasing their robustness.

We experiment with three different NMT systems with access to character information at different levels. First, we use the fully character-level model of Lee et al. (2017). This is a sequence-to-sequence model with attention (Sutskever et al., 2014; Bahdanau et al., 2014) that is trained on characters to characters (char2char). It has a complex encoder with convolutional, highway, and recurrent layers, and a standard recurrent decoder. See Lee et al. (2017) for architecture details. This model was shown to have excellent performance on the German→English and Czech→English language pairs. We use the pre-trained German/Czech→English models.

Second, we use Nematus (Sennrich et al., 2017), a popular NMT toolkit that was used in top-performing contributions in shared MT tasks in WMT (Sennrich et al., 2016b) and IWSLT (Junczys-Dowmunt & Birch, 2016). It is another sequence-to-sequence model with several architecture modifications, especially operating on sub-word units using byte-pair encoding (BPE) (Sennrich et al., 2016a). We experimented with both their single best and ensemble BPE models, but saw no significant difference in their performance under noise, so we report results with their single best WMT models for German/Czech→English.

Finally, we train an attentional sequence-to-sequence model with a word representation based on a character convolutional neural network (charCNN). This model retains the notion of a word but learns a character-dependent representation of words. It was shown to perform well on morphologically-rich languages (Kim et al., 2015; Belinkov & Glass, 2016; Costa-jussà & Fonollosa, 2016; Sajjad et al., 2017), thanks to its ability to learn morphologically-informative representations (Belinkov et al., 2017a). The charCNN model has two long short-term memory (Hochreiter & Schmidhuber, 1997) layers in the encoder and decoder. A CNN over characters in each word replaces the word embeddings on the encoder side (for simplicity, the decoder is word-based). We use 1000 filters with a width of 6 characters. The character embedding size is set to 25. The convolutions are followed by Tanh and max-pooling over the length of the word (Kim et al., 2015). We train charCNN with the implementation in Kim (2016); all other settings are kept to default values.

## 4  Data

### 4.1  MT Data

We use the TED talks parallel corpus prepared for IWSLT 2016 (Cettolo et al., 2012) for testing all of the NMT systems, as well as for training the charCNN models. We follow the official training/development/test splits. All texts are tokenized with the Moses tokenizer. Table 1 summarizes statistics on the TED talks corpus.

Table 1: Statistics for the source-side of French/German/Czech→English parallel corpora.

|  | French | | | German | | | Czech | | |
| --- | --- | --- | --- | --- | --- | --- | --- | --- | --- |
|  | Train | Dev | Test | Train | Dev | Test | Train | Dev | Test |
| Sentences | 235K | 2.5K | 0.8K | 210K | 2.5K | 1.4K | 122K | 20K | 1K |
| Words | 5.2M | 55K | 16K | 4M | 50K | 26K | 2.1M | 35K | 15K |

Table 2: Average number of available edits per word in natural noise datasets and the corresponding token recall of those edits on the training and test splits.

| French | | | | German | | | | Czech | | | |
| --- | --- | --- | --- | --- | --- | --- | --- | --- | --- | --- | --- |
| Words | Errors | Train | Test | Words | Errors | Train | Test | Words | Errors | Train | Test |
| 65,156 | 2.7 | 40% | 41% | 1,344 | 2.5 | 37% | 40% | 6,036 | 2.6 | 46% | 51% |

## 4.2 Noise: Natural and Artificial

We insert noise into the source-side of the parallel MT data by utilizing naturally occurring errors and generating synthetic ones. In order to facilitate future work on noise in NMT, we release code and data for generating the noise used in our experiments.[3]

### 4.2.1 Natural noise

Since we do not have access to a parallel corpus with natural noise, we instead harvest naturally occurring errors (typos, misspellings, etc.) from available corpora of edits to build a look-up table of possible lexical replacements. In this work, we restrict ourselves to single word replacements, but several of the corpora below also provide access to phrase replacements.

**French**  Max & Wisniewski (2010) collected Wikipedia edit histories to form the Wikipedia Correction and Paraphrase Corpus (WiCoPaCo). They found the bulk of edits were due to incorrect diacritics, choosing the wrong homophone, and incorrect grammatical conjugation.

**German**  Our German data combines two projects: RWSE Wikipedia Revision Dataset (Zesch, 2012) and The MERLIN corpus of language learners (Wisniewski et al., 2013). These corpora were created to measure spelling difficulty and test models of contextual fitness. Unfortunately, the datasets are quite small so we have combined them here.

**Czech**  Our Czech errors come from manually annotated essays written by non-native speakers (Šebesta et al., 2017). Here, the authors found an incredibly diverse set of errors, and therefore phenomena of interest: capitalization, incorrectly replacing voiced and voiceless consonants (e.g. z/s, g/k), missing palatalization (matĕe/matce), error in valence, pronominal reference, inflection, colloquial forms, and so forth. Their analysis gives us the best insight into how difficult it would be to synthetically generate truly natural errors. We found similarly rich errors in German (Section 7.2).

We insert these errors into the source-side of the parallel data by replacing every word in the corpus with an error if one exists in our dataset. When there is more than one possible replacement to choose we sample uniformly. Words for which there is no error are kept as is. Table 2 shows the number of words for which we were able to collect errors in each language, and the average number of errors per word. Despite the small size of the German and Czech datasets, we are able to replace up to half of the words in the corpus with errors. Due to the small size of the German and Czech datasets these percentages decrease for longer words ($> 4$ characters) to 25% and 32%, respectively.

### 4.2.2 Synthetic noise

In addition to naturally collected sources of error, we also experiment with four types of synthetic noise: Swap, Middle Random, Fully Random, and Keyboard Typo.

---

[3] https://github.com/ybisk/charNMT-noise

Table 3: The effect of Natural (`Nat`) and synthetic noise (Swap `swap`, Middle Random `Mid`, Fully Random `Rand`, and Keyboard Typo `Key`) on models trained on clean (Vanilla) texts.

| | | | Synthetic | | | | |
|---|---|---|---|---|---|---|---|
| | | Vanilla | Swap | Mid | Rand | Key | Nat |
| French | charCNN | 42.54 | 10.52 | 9.71 | 1.71 | 8.26 | 17.42 |
| German | charCNN | 34.79 | 9.25 | 8.37 | 1.02 | 6.40 | 14.02 |
| | char2char | 29.97 | 5.68 | 5.46 | 0.28 | 2.96 | 12.68 |
| | Nematus | 34.22 | 3.39 | 5.16 | 0.29 | 0.61 | 10.68 |
| Czech | charCNN | 25.99 | 6.56 | 6.67 | 1.50 | 7.13 | 10.20 |
| | char2char | 25.71 | 3.90 | 4.24 | 0.25 | 2.88 | 11.42 |
| | Nematus | 29.65 | 2.94 | 4.09 | 0.66 | 1.41 | 11.88 |

Table 4: An example noisy text with human and machine translations.

| | |
|---|---|
| Input | Luat eienr Stduie der Cambrdige Unievrstit speilt es kenie Rlloe in welcehr Reiehnfogle die Buhcstbaen in eniem Wrot vorkmomen, die eingzie whctige Sahce ist, dsas der ertse und der lettze Buhcstbaen stmimt . |
| Human | According to a study from Cambridge university, it doesn't matter which order letters in a word are, the only important thing is that the first and the last letter appear in their correct place. |
| char2char | Cambridge Universttte is one of the most important features of the Cambridge Universttten , which is one of the most important features of the Cambridge Universttten . |
| Nematus | Luat eienr Stduie der Cambrant Unievrstilt splashed it kenie Rlloe in welcehr Reiehnfogle the Buhcstbaen in eniem Wred vorkmomen, die eingzie whcene Sahce ist, DSAs der ertse und der lettze Buhcstbaen stmimt . |
| charCNN | According to the <unk> of the Cambridge University , it 's a little bit of crude oil in a little bit of recycling , which is a little bit of a cool cap , which is a little bit of a strong cap , that the fat and the <unk> bites is consistent . |

**Swap : `Swap`** The simplest source of noise is swapping two letters (e.g. *noise→nosie*). This is common when typing quickly and is easily implemented. We perform one swap per word, but do not alter the first or last letters. For this reason, this noise is only applied to words of length $\geq 4$.

**Middle Random : `Mid`** Following the claims of the previously discussed meme, we randomize the order of all the letters in a word except for the first and last (*noise→nisoe*). Again, by necessity, this means we do not alter words shorter than four characters.

**Fully Random : `Rand`** As we are unaware of any strong results on the importance of the first and last letters we also include completely randomized words (*noise→iones*). This is a particularly extreme case, but we include it for completeness. This type of noise is applied to all words.

**Keyboard Typo : `Key`** Finally, using the traditional keyboards for our languages, we randomly replace one letter in each word with an adjacent key (*noise→noide*). This type of error should be much easier than the random settings as most of the word is left intact, but does introduce a completely new character which will often break the templates a system has learned to rely on.

## 5 FAILURES TO TRANSLATE NOISY TEXTS

Table 3 shows BLEU scores of models trained on clean (Vanilla) texts and tested on clean and noisy texts. All models suffer a significant drop in BLEU when evaluated on noisy texts. This is true for both natural noise and all kinds of synthetic noise. The more noise in the text, the worse the translation quality, with random scrambling producing the lowest BLEU scores.

The degradation in translation quality is especially severe in light of humans' ability to understand noisy texts. To illustrate this, consider the noisy text in Table 4. Humans are quite good at understanding such scrambled texts in a variety of languages.[4] We also verified this by obtaining a

---

[4]Matt Davis has a wide collection of translations of this text in multiple languages: https://www.mrc-cbu.cam.ac.uk/personal/matt.davis/Cmabrigde/.

Table 5: Google Translate's performance with natural errors and the gains from using spell checking.

| French | | | German | | | Czech | | |
|---|---|---|---|---|---|---|---|---|
| Vanilla | Nat | Spelling | Vanilla | Nat | Spelling | Vanilla | Nat | Spelling |
| 43.3 | 16.7 | 21.4 | 38.7 | 18.6 | 25.0 | 26.5 | 12.3 | 11.2 |

Table 6: Results of `meanChar` models trained and tested on different noise conditions: Scrambled (Scr), Keyboard Typo (`Key`), and Natural (`Nat`).

| Test \ Train | French | | | German | | | Czech | | |
|---|---|---|---|---|---|---|---|---|---|
| | Scr | Key | Nat | Scr | Key | Nat | Scr | Key | Nat |
| Vanilla | 34.26 | 4.27 | 12.58 | 27.53 | 3.34 | 9.41 | 3.73 | 2.06 | 3.25 |
| Key | 31.88 | 29.75 | 13.16 | 10.04 | 8.84 | 4.45 | 2.03 | 1.9 | 1.42 |
| Nat | 26.94 | 5.30 | 27.49 | 15.65 | 3.06 | 26.26 | 1.66 | 1.52 | 1.58 |
| Rand + Key | 13.60 | 11.09 | 6.12 | 26.59 | 22.41 | 11.07 | 9.97 | 7.48 | 4.21 |
| Rand + Nat | 28.28 | 5.10 | 20.40 | 13.87 | 3.73 | 12.74 | 4.89 | 2.82 | 3.42 |
| Key + Nat | 31.30 | 26.94 | 24.24 | 6.62 | 5.41 | 5.75 | 1.62 | 1.68 | 1.58 |
| Rand + Key + Nat | 3.10 | 3.28 | 2.76 | 8.02 | 5.79 | 6.36 | 1.73 | 1.74 | 1.66 |

translation from a German native-speaker, unfamiliar with the meme. As shown in the table, the speaker had no trouble understanding and translating the sentence properly. In contrast, the state-of-the-art systems (`char2char` and `Nematus`) fail on this text.

One natural question is if robust spell checkers trained on human errors are sufficient to address this performance gap. To test this, we ran texts with and without natural errors through Google Translate. We then used Google's spell-checkers to correct the documents. We simply accepted the first suggestion for every detected mistake detected, and report results in Table 5.

We found that in French and German, there was often only a single predicted correction and this corresponds to roughly +5 or more in BLEU. In Czech, however, there was often a large list of possible conjugations and changes, likely indicating that a rich grammatical model would be necessary to predict the correction. It is also important to note the substantial drops from vanilla text even with spell check. This suggests that natural noise cannot be easily addressed by existing tools.

## 6 DEALING WITH NOISE

### 6.1 STRUCTURE INVARIANT REPRESENTATIONS

The three NMT models are all sensitive to word structure. The `char2char` and `charCNN` models both have convolutional layers on character sequences, designed to capture character n-grams. The model in `Nematus` is based on sub-word units obtained with BPE. It thus relies on character order within and across sub-word units. All these models are therefore sensitive to types of noise generated by character scrambling (`Swap`, `Mid`, and `Rand`). Can we improve model robustness by adding invariance to these kinds of noise? Perhaps the simplest such model is to take the average character embedding as a word representation. This model, referred to as `meanChar`, first generates a word representation by averaging character embeddings, and then proceeds with a word-level encoder similar to the `charCNN` model. The `meanChar` model is by definition insensitive to scrambling, although it is still sensitive to other kinds of noise (`Key` and `Nat`).

Table 6 (first row) shows the results of `meanChar` models trained on vanilla texts and tested on noisy texts (the results on vanilla texts are by definition equal to those on scrambled texts). Overall, the average character embedding proves to be a pretty good representation for translating scrambled texts: while performance drops by about 7 BLEU points below `charCNN` on vanilla French and German, it is much better than `charCNN`'s performance on scrambled texts (compare to Table 3). The results of `meanChar` on Czech are much worse, possibly due to its more complex morphology. However, the `meanChar` model performance degrades quickly on other kinds of noise as the model trained on vanilla texts was not designed to handle `Nat` and `Key` types of noise.

Table 7: Results of `charCNN` models trained and tested on different noise conditions.

| | Test / Train | Vanilla | Swap | Mid | Rand | Key | Nat | Ave |
|---|---|---|---|---|---|---|---|---|
| French | Swap | 39.01 | **42.56** | 33.64 | 2.72 | 4.85 | 16.43 | 23.20 |
| | Mid | **42.46** | 42.19 | **42.17** | 3.36 | 6.20 | 18.22 | 25.77 |
| | Rand | 39.53 | 39.46 | 39.13 | **39.73** | 3.11 | 16.63 | 29.60 |
| | Key | 38.49 | 10.56 | 8.69 | 1.08 | **38.88** | 16.86 | 19.10 |
| | Nat | 28.77 | 12.45 | 8.39 | 1.03 | 6.61 | **36.00** | 15.54 |
| | Rand + Key | 39.23 | 38.85 | 38.89 | 39.13 | 38.22 | 18.71 | 35.51 |
| | Rand + Nat | 36.86 | 38.95 | 38.44 | 38.63 | 6.67 | 33.89 | 32.24 |
| | Key + Nat | 38.47 | 17.33 | 10.54 | 1.52 | 38.62 | 34.66 | 23.52 |
| | Rand + Key + Nat | 36.97 | 36.92 | 36.65 | 36.64 | 35.25 | 31.77 | **35.70** |
| German | Swap | 32.66 | **34.76** | 29.03 | 2.19 | 4.78 | 13.37 | 19.47 |
| | Mid | **34.32** | 34.26 | **34.27** | 3.50 | 5.08 | 14.43 | 20.98 |
| | Rand | 33.65 | 33.44 | **33.75** | 33.56 | 3.00 | 14.47 | 25.31 |
| | Key | 32.87 | 10.13 | 8.39 | 1.16 | **33.28** | 13.88 | 16.62 |
| | Nat | 25.79 | 8.20 | 5.73 | 0.93 | 4.80 | **34.59** | 13.34 |
| | Rand + Key | 32.03 | 31.57 | 31.32 | 31.58 | 31.23 | 15.59 | 28.89 |
| | Rand + Nat | 32.37 | 32.40 | 31.91 | 32.11 | 4.77 | 33.00 | 27.76 |
| | Key + Nat | 30.39 | 13.51 | 8.99 | 1.53 | 32.23 | 33.46 | 20.02 |
| | Rand + Key + Nat | 31.29 | 30.93 | 30.54 | 30.04 | 29.81 | 31.60 | **30.70** |
| Czech | Swap | **24.22** | **24.90** | 18.72 | 2.72 | 6.00 | 9.03 | 14.27 |
| | Mid | 23.81 | **24.52** | 24.08 | 3.96 | 6.34 | 9.54 | 15.38 |
| | Rand | 23.44 | 23.31 | 23.24 | **23.47** | 3.70 | 8.10 | 17.54 |
| | Key | 23.15 | 7.06 | 6.04 | 1.56 | **22.80** | 10.16 | 11.80 |
| | Nat | 18.04 | 5.36 | 4.48 | 1.47 | 6.71 | **21.64** | 9.62 |
| | Rand + Key | 21.46 | 20.81 | 20.90 | 20.59 | 19.48 | 8.72 | 18.66 |
| | Rand + Nat | 20.59 | 21.56 | 20.49 | 20.53 | 5.89 | 18.39 | 17.91 |
| | Key + Nat | 19.55 | 6.59 | 5.72 | 1.40 | 21.31 | 19.54 | 12.35 |
| | Rand + Key + Nat | 21.30 | 21.33 | 20.38 | 19.94 | 19.25 | 18.38 | **20.10** |

## 6.2 BLACK-BOX ADVERSARIAL TRAINING

To increase model robustness we follow a black-box adversarial training scenario, where the model is presented with adversarial examples that are generated without direct access to the model (Papernot et al., 2016a; 2017; Liu et al., 2017; Narodytska & Kasiviswanathan, 2017; Jia & Liang, 2017). We replace the original training set with a noisy training set, where noise is introduced according to the description in Section 4.2. The noisy training set has exactly the same number of sentences and words as the training set. We have one fixed noisy training set per each noise type.[5]

As shown in Table 6 (second block), training on noisy text can lead to improved performance. The `meanChar` models trained on `Key` perform well on `Key` in French, but not in the other languages. The models trained on `Nat` perform well in French and German, but not in Czech. Overall, training the `meanChar` model on noisy text does not appear to consistently increase its robustness to different kinds of noise. The `meanChar` model however was not expected to perform well on nonscrambling types of noise. Next we test whether the more complicated `charCNN` model is more robust to different kinds of noise, by training on noisy texts. The results are shown in Table 7.

In general, `charCNN` models that are trained on a specific kind of noise perform well on the same kind of noise at test time (results in **bold**). All models also maintain a fairly good quality on vanilla texts. The robust training is sensitive to the kind of noise. Among the scrambling methods (`Swap`/`Mid`/`Rand`), more noise helps in training: models trained on `random` noise can still translate `Swap`/`Mid` noise, but not vice versa. The three broad classes of noise (scrambling, `Key`, `Nat`)

---

[5]When replacing words in the input, we inevitably make some of the same replacements on both the training and test sets. We verify this does not decrease the percent of unseen words in testing. Conversely, we found it increases for all synthetic noise types and is similar for the vanilla and natural noise conditions.

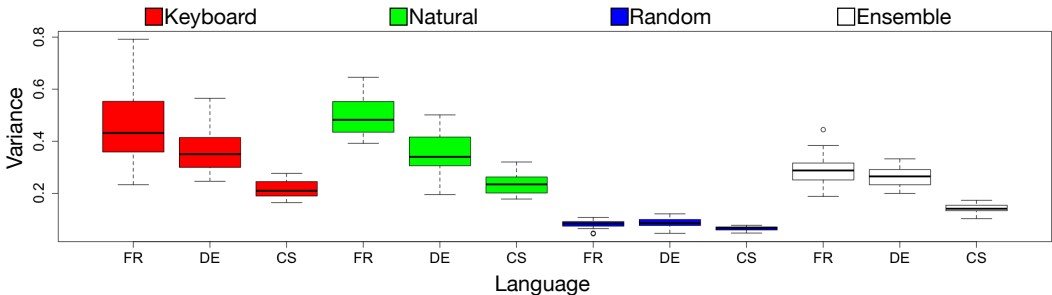

Figure 2: Variances of `charCNN` weights when trained on only `Key`, `Natural`, `Random` noise and on a mix of all three are shown in red, green, blue, and white, respectively

are not mutually-beneficial. Models trained on one do not perform well on the others. In particular, only models trained on natural noise can reasonably translate natural noise at test time. We find this result indicates an important difference between computational models and human performance, since humans can decipher random letter orderings without explicit training of this form.

Next, we test whether we can increase training robustness by exposing the model to multiple types of noise during training. Our motivation is to see if models can perform well on more than one kind of noise. We therefore mix up to three kinds of noise by sampling a noise method uniformly at random for each sentence. We then train a model on the mixed noisy training set and test it on both vanilla and (unmixed) noisy versions of the test set. We find that models trained on mixed noise are slightly worse than models trained on unmixed noise. However, the models trained on mixed noise are robust to the specific types of noise they were trained on. In particular, the model trained on a mix of `Rand`, `Key`, and `Nat` noise is robust to all noise kinds. Even though it is not the best on any one kind of noise, it achieves the best result on average.

This model is also able to translate the scrambled meme reasonably well:

> "According to a study of Cambridge University, it doesn't matter which technology in a word is going to get the letters in a word that is the only important thing for the first and last letter."

## 7 ANALYSIS

### 7.1 LEARNING MULTIPLE KINDS OF NOISE IN `CHARCNN`

The `charCNN` model was able to perform well on all kinds of noise by training on a mix of noise types. In particular, it performed well on scrambled characters even though its convolutions should be sensitive to the character order, as opposed to `meanChar` which is by definition invariant to character order. How then can `charCNN` learn to be robust to multiple kinds of noise at the same time? We speculate that different convolutional filters learn to be robust to different kinds of noise. A convolutional filter can in principle capture a mean (or sum) operation by employing equal or close to equal weights.

To test this, we analyze the weights learned by `charCNN` models trained under four conditions: three models trained each on completely scrambled words (`Rand`), keyboard typos (`Key`), and natural human errors (`Nat`), as well as an ensemble model trained on a mix of `Rand+Key+Nat` kinds of noise. For each model, we compute the variance across the filter width (6 characters) for each one of the 1000 filters and for each one out of 25 character embedding dimensions. Intuitively, this variance captures how much a particular filter learns a uniform vs. non-uniform combination of characters. Then we average the variances across the 1000 filters. This yields 25 averaged variances, one for each character embedding dimension. Low average variance means that different filters tend to learn similar behaviors, while high average variance means that they learn different patterns.

Figure 2 shows a box plot of these averages for our three languages and four training conditions. Clearly, the variances of the weights learned by the `Rand` model are much smaller than those of the weights learned by any other setting. This makes sense as with random scrambling there are no

patterns to detect in the data, so filters resort to close to uniform weights. In contrast, the `Key` and `Nat` settings introduce a large set of new patterns for the CNNs to try and learn, leading to high variances. Finally, the ensemble model trained on mixed noise appears to be in the middle as it tries to capture both the uniform relationships of `Rand` and the more diverse patterns of `Nat + Key`.

Moreover, the variance of variances (size of the box) is smallest in the `Rand` setting, larger in the mixed noise model, and largest in `Key` and `Nat`. This indicates that filters for different character embedding dimensions are more different from one another in `Key` and `Nat` models. In contrast, in the `Rand` model, the variance of variances is close to zero, indicating that in all character embedding dimensions the learned weights are of small variance; they do similar things, that is, the model learned to reproduce a representation similar to the `meanChar` model. The ensemble model again seems to find a balance between `Rand` and `Key`/`Nat`.

## 7.2 RICHNESS OF NATURAL NOISE

Natural noise appears to be very different from synthetic noise. None of the models that were trained only on synthetic noise were able to perform well on natural noise. We manually analyzed a small sample (~40 examples) of natural noise from the German dataset. We found that the most common sources of noise are phonetic or phonological phenomena in the language (34%) and character omissions (32%). The rest are incorrect morphological conjugations of verbs, key swaps, character insertions, orthographic variants, and other errors. Table 8 shows examples of these kinds of noise.

The most common types of natural noise – phonological and omissions – are not directly captured by our synthetic noise generation, and demonstrate that good synthetic errors will likely require more explicit phonemic and linguistic knowledge. This discrepancy helps explain why the models trained on synthetic noise were not particularly successful in translating natural noise.

Table 8: Examples of natural noise from the German errors dataset.

| Error type | Examples |
|---|---|
| Phonetic | Tut/Tud (devoicing of final stops), sieht/zieht (s = /z/ before vowel), Trotzdem/Trozdem (tz = /z/), gekriegt/gekrigt (vowel length), Natürlich/Naturlich/Näturlich (diacritics) |
| Omission | erfahren/erfaren, Babysitter/Babysiter, selbst/sebst, Hausschuhe/Hausschue |
| Morphological | wohnt/wonnen, fortsetzt/forzusetzen, wünsche/wünchen |
| Key swap | Eltern/Eltren, Deine/Diene, nichts/nichst, Bahn/Bhan |
| Other | Agglomerationen/Agromelationen (omission + letter swap), Hausaufgabe/Hausausgabe, Thema/Temer, Detailhandelsfachfrau/Deitellhandfachfrau |

## 8 CONCLUSION

In this work, we have shown that character-based NMT models are extremely brittle and tend to break when presented with both natural and synthetic kinds of noise. We investigated methods for increasing their robustness by using a structure-invariant word representation and by ensemble training on adversarial examples of different kinds. We found that a character-based CNN can learn to address multiple types of errors that are seen in training. However, we observed rich characteristics of natural human errors that cannot be easily captured by existing models. Future work might investigate using phonetic and syntactic structure to generate more realistic synthetic noise.

We believe that more work is necessary in order to immune NMT models against natural noise. As corpora with natural noise are limited, another approach to future work is to design better NMT architectures that would be robust to noise without seeing it in the training data. New psychology results on how humans cope with natural noise might point to possible solutions to this problem.

## ACKNOWLEDGEMENTS

This work benefited from discussions with Frank Keller. This work was supported by the Qatar Computing Research Institute (QCRI) and Samsung Research.

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
