# OpenReview forum: "Synthetic and Natural Noise Both Break Neural Machine Translation"
_ICLR.cc/2018/Conference — Accept (Oral)_

### Official Review · AnonReviewer1 · 2017-11-27
**Nice empirical paper on robustness of NMT**

**Rating:** 7
**Confidence:** 4

**Review:**

This paper empirically investigates the performance of character-level NMT systems in the face of character-level noise, both synthesized and natural. The results are not surprising:

* NMT is terrible with noise.

* But it improves on each noise type when it is trained on that noise type.

What I like about this paper is that:

1) The experiments are very carefully designed and thorough.

2) This problem might actually matter. Out of curiosity, I ran the example (Table 4) through Google Translate, and the result was gibberish. But as the paper shows, it’s easy to make NMT robust to this kind of noise, and Google (and other NMT providers) could do this tomorrow. So this paper could have real-world impact.

3) Most importantly, it shows that NMT’s handling of natural noise does *not* improve when trained with synthetic noise; that is, the character of natural noise is very different. So solving the problem of natural noise is not so simple… it’s a *real* problem. Speculating, again: commercial MT providers have access to exactly the kind of natural spelling correction data that the researchers use in this paper, but at much larger scale. So these methods could be applied in the real world. (It would be excellent if an outcome of this paper was that commercial MT providers answered it’s call to provide more realistic noise by actually providing examples.)

There are no fancy new methods or state-of-the-art numbers in this paper. But it’s careful, curiosity-driven empirical research of the type that matters, and it should be in ICLR.

---

> ### Author Response · Authors · 2017-12-10
> **We would love to see real world impact and new data collection**
>
> Thank you for the useful feedback.
>
> 1. We agree that the topic has real-world impact for MT providers and will emphasize this in the conclusions.
>
> 2. We would love to see MT providers use noisy data and we agree that the community would benefit from access to more noisy examples.

---

### Official Review · AnonReviewer2 · 2017-11-27
**Good work on an under-studied problem**

**Rating:** 7
**Confidence:** 4

**Review:**

This paper investigates the impact of character-level noise on various flavours of neural machine translation. It tests 4 different NMT systems with varying degrees and types of character awareness, including a novel meanChar system that uses averaged unigram character embeddings as word representations on the source side. The authors test these systems under a variety of noise conditions, including synthetic scrambling and keyboard replacements, as well as natural (human-made) errors found in other corpora and transplanted to the training and/or testing bitext via replacement tables. They show that all NMT systems, whether BPE or character-based, degrade drastically in quality in the presence of both synthetic and natural noise, and that it is possible to train a system to be resistant to these types of noise by including them in the training data. Unfortunately, they are not able to show any types of synthetic noise helping address natural noise. However, they are able to show that a system trained on a mixture of error types is able to perform adequately on all types of noise.

This is a thorough exploration of a mostly under-studied problem. The paper is well-written and easy to follow. The authors do a good job of positioning their study with respect to related work on black-box adversarial techniques, but overall, by working on the topic of noisy input data at all, they are guaranteed novelty. The inclusion of so many character-based systems is very nice, but it is the inclusion of natural sources of noise that really makes the paper work. Their transplanting of errors from other corpora is a good solution to the problem, and one likely to be built upon by others. In terms of negatives, it feels like this work is just starting to scratch the surface of noise in NMT. The proposed meanChar architecture doesn’t look like a particularly good approach to producing noise-resistant translation systems, and the alternative solution of training on data where noise has been introduced through replacement tables isn’t extremely satisfying. Furthermore, the use of these replacement tables means that even when the noise is natural, it’s still kind of artificial. Finally, this paper doesn’t seem to be a perfect fit for ICLR, as it is mostly experimental with few technical contributions that are likely to be impactful; it feels like it might be more at home and have greater impact in a *ACL conference.

Regarding the artificialness of their natural noise - obviously the only solution here is to find genuinely noisy parallel data, but even granting that such a resource does not yet exist, what is described here feels unnaturally artificial. First of all, errors learned from the noisy data sources are constrained to exist within a word. This tilts the comparison in favour of architectures that retain word boundaries (such as the charCNN system here), while those systems may struggle with other sources of errors such as missing spaces between words. Second, if I understand correctly, once an error is learned from the noisy data, it is applied uniformly and consistently throughout the training and/or test data. This seems worse than estimating the frequency of the error and applying them stochastically (or trying to learn when an error is likely to occur). I feel like these issues should at least be mentioned in the paper, so it is clear to the reader that there is work left to be done in evaluating the system on truly natural noise.

Also, it is somewhat jarring that only the charCNN approach is included in the experiments with noisy training data (Table 6). I realize that this is likely due to computational or time constraints, but it is worth providing some explanation in the text for why the experiments were conducted in this manner. On a related note, the line in the abstract stating that “... a character convolutional neural network  is able to simultaneously learn representations robust to multiple kinds of noise” implies that the other (non-charCNN) architectures could not learn these representations, when in reality, they simply weren’t given the chance.

Section 7.2 on the richness of natural noise is extremely interesting, but maybe less so to an ICLR audience. From my perspective, it would be interesting to see that section expanded, or used as the basis for future work on improve architectures or training strategies.

I have only one small, specific suggestion: at the end of Section 3, consider deleting the last paragraph break, so there is one paragraph for each system (charCNN currently has two paragraphs).

[edited for typos]

---

> ### Author Response · Authors · 2017-12-10
> **Thank you for your detailed comments**
>
> Thank you for the useful feedback.  We agree that noisy input in neural machine translation is an under-studied problem.
>
> Responses to specific comments:
> 1. We agree that our work only starts to scratch the surface of noise in NMT and believe there’s much more to be done in this area. We do believe that it’s important to initiate a discussion of this issue in the ICLR community, for several reasons: (a) we study word and character representations for NMT, which is in line with the ICLR representation learning theme; (b) ICLR audience is very interested in neural machine translation and seminal work on NMT has been published in ICLR (e.g., Bahdanau et al.’s 2015 paper on attention in NMT); (c) ICLR audience is very interested in noise and adversarial examples, as evidenced by the plethora of recent papers on the topic. As reviewer 1 says, even though there are no fancy new methods in the paper, we believe that this kind of research belongs in ICLR.
>
> 2. We agree that meanChar may not be the ideal architecture for capturing noise, but it’s a simple, structure-invariant representation that works reasonably well. We have tried several other architectures, including a self-attention mechanism, but haven’t been able to improve beyond it. We welcome more suggestions and can include those negative results in new drafts of the paper.
>
> 3. Training with noise has its limitations, but it’s an effective method that can be employed by NMT providers and researchers easily and impactfully, as pointed out by reviewer 1.
>
> 4. In this work, we focus on word-level noise. Certainly, sentence-level noise is also important to learn, and we’d like to see more work on this. We’ll add this as another direction for future work. Note that while charCNN may have some advantage in dealing with word-level noise, it too suffers from increasing amounts of noise, similar to the other models we studied.
>
> 5. Applying noise stochastically based on frequency in available corpora is an interesting suggestion, that can be done for the natural noise, but not so clear how to apply for synthetic noise. We did experiment with increasing amounts of noise (Figure 1), but we agree there’s more to be done. We’ll add this as another future work.
>
> 6. (To both reviewer 2 and 3) Regarding training other seq2seq models with noise: Our original intent was to test the robustness of pre-trained state-of-the-art models, but we also considered retraining them in this noisy paradigm. There are a number of design decisions that are involved here (e.g. should the BPE dictionary be built on the noisy texts and how should thresholds be varied?). That being said, we can investigate training using published parameter values, but worry these may be wholly inappropriate settings for the new noisy data.
>
> 7. We’ll modify the abstract to not give the wrong impression regarding what other architectures can learn.
>
> 8. We included section 7.2 to demonstrate why synthetic noise is not very helpful in dealing with natural noise, as well as to motivate the development of better architectures.
>
> 9. We’ll correct the other small issues pointed to.

---

> > ### Comment · AnonReviewer2 · 2018-01-12
> > **Thanks!**
> >
> > Thanks for your thoughtful response to my review.

---

### Official Review · AnonReviewer3 · 2017-11-29
**Interesting study of the impact of noisy text on MT quality**

**Rating:** 8
**Confidence:** 4

**Review:**

This paper investigates the impact of noisy input on Machine Translation, and tests simple ways to make NMT models more robust.

Overall the paper is a clearly written, well described report of several experiments. It shows convincingly that standard NMT models completely break down on both natural "noise" and various types of input perturbations. It then tests how the addition of noise in the input helps robustify the charCNN model somewhat. The extent of the experiments is quite impressive: three different NMT models are tried, and one is used in extensive experiments with various noise combinations.

This study clearly addresses an important issue in NMT and will be of interest to many in the NLP community. The outcome is not entirely surprising (noise hurts and training and the right kind of noise helps) but the impact may be. I wonder if you could put this in the context of "training with input noise", which has been studied in Neural Network for a while (at least since the 1990s). I.e. it could be that each type of noise has a different regularizing effect, and clarifying what these regularizers are may help understand the impact of the various types of noise. Also, the bit of analysis in Sections 6.1 and 7.1 is promising, if maybe not so conclusive yet.

A few constructive criticisms:

The way noise is included in training (sec. 6.2) could be clarified (unless I missed it) e.g. are you generating a fixed "noisy" training set and adding that to clean data? Or introducing noise "on-line" as part of the training? If fixed, what sizes were tried? More information on the experimental design would help.

Table 6 is highly suspect: Some numbers seem to have been copy-pasted in the wrong cells, eg. the "Rand" line for German, or the Swap/Mid/Rand lines for Czech. It's highly unlikely that training on noisy Swap data would yield a boost of +18 BLEU points on Czech -- or you have clearly found a magical way to improve performance.

Although the amount of experiment is already important, it may be interesting to check whether all se2seq models react similarly to training with noise: it could be that some architecture are easier/harder to robustify in this basic way.

[Response read -- thanks]
I agree with authors that this paper is suitable for ICLR, although it will clearly be of interest to ACL/MT-minded folks.

---

> ### Author Response · Authors · 2017-12-10
> **Thank you for your insights and catching our copy/paste error.**
>
> Thank you for the constructive feedback.
> 1. Noise setup: when training with noise, we replace the original training set with a new, noisy training set. The noisy training set has exactly the same number of sentences and words as the training set, but noise is introduced according to the description in Section 4. Therefore, we have one fixed noisy training set per each noise type. We’ll clarify the experimental design in the paper.
>
> 2. We had not thought to explore the relationship between the noise we are introducing as a corruption of the input and the training under noise paradigm you referenced. We might be mistaken, but normally, the corruption (e.g. Bishop 95) is in the form of small additive gaussian noise. It isn’t immediately clear to us whether discrete perturbation of the input like we have here is equivalent, but would love suggestions on analyses we might do to investigate this insight further.
>
> 3. Some cells in the mentioned rows in Table 6 were indeed copied from the French rows by error. We corrected the numbers and they are in line with the overall trends. Thank you for pointing this out.  The corrected Czech numbers are in the 20s and the best performing system is the Rand+Key+Real setting.
>
> 4. (To both reviewer 2 and 3) Regarding training other seq2seq models with noise: Our original intent was to test the robustness of pre-trained state-of-the-art models, but we also considered retraining them in this noisy paradigm. There are a number of design decisions that are involved here (e.g. should the BPE dictionary be built on the noisy texts and how should thresholds be varied?). That being said, we can investigate training using published parameter values, but worry these may be wholly inappropriate settings for the new noisy data.

---

### Public Comment · (anonymous) · 2017-12-05
**Better suited to NLP conference**

The paper points out the lack of robustness of character based models and explores a few, very basic solutions, none of which are effective. While starting a discussion around this problem is valuable, the paper provides no actually working solutions, and the solutions explored are very basic from a machine learning point of view. This publication is better suited to a traditional NLP venue such as ACL/EMNLP.

---

> ### Author Response · Authors · 2017-12-10
> **Thank you for your feedback**
>
> 1. We believe that the topic on noise in NMT is of interest to the ICLR audience. Please see our response to reviewer 1 for a detailed explanation.
>
> 2. We find that both solutions we offered are effective to a reasonable extent. meanChar works fairly well on scrambling types of noise, but fails on other noise, as expected. Adversarial training with noise works well as long as train/test noise types are matched, so it’s a useful practical technique that can be applied in NMT systems, as pointed out by reviewer 1.

---

> ### Comment · AnonReviewer1 · 2018-01-11
> **Strange comment**
>
> The CFP clearly states that "applications in vision, audio, speech, natural language processing, robotics, neuroscience, or any other field" are relevant.

---

### Decision · Program_Chairs · 2018-01-29
**ICLR 2018 Conference Acceptance Decision**

**Decision:**

Accept (Oral)

**Comment:**

The pros and cons of this paper cited by the reviewers can be summarized below:

Pros:
* The paper is a first attempt to investigate an under-studied area in neural MT (and potentially other applications of sequence-to-sequence models as well)
* This area might have a large impact; existing models such as Google Translate fail badly on the inputs described here
* Experiments are very carefully designed and thorough
* Experiments on not only synthetic but also natural noise add significant reliability to the results
* Paper is well-written and easy to follow

Cons:
* There may be better architectures for this problem than the ones proposed here
* Even the natural noise is not entirely natural, e.g. artificially constrained to exist within words
* Paper is not a perfect fit to ICLR (although ICLR is attempting to cast a wide net, so this alone is not a critical criticism of the paper)

This paper had uniformly positive reviews and has potential for large real-world impact.